# When Two plus Two Is More than Four: Evidence for a Synergistic Effect of Fatty Acids on Peroxisome Proliferator—Activated Receptor Activity in a Bovine Hepatic Model

**DOI:** 10.3390/genes12081283

**Published:** 2021-08-21

**Authors:** Sebastiano Busato, Massimo Bionaz

**Affiliations:** Department of Animal and Rangeland Sciences, Oregon State University, Corvallis, OR 97330, USA; busatos@oregonstate.edu

**Keywords:** PPAR, ruminant nutrigenomics, gene–nutrient interactions

## Abstract

The inclusion of fat in livestock diets represents a valuable and cost-effective way to increase the animal’s caloric intake. Beyond their caloric value, fatty acids can be understood in terms of their bioactivity, via the modulation of the ligand-dependent nuclear peroxisome proliferator-activated receptors (PPAR). Isotypes of PPAR regulate important metabolic processes in both monogastric and ruminant animals, including the metabolism of fatty acids (FA), the production of milk fat, and the immune response; however, information on the modulation of bovine PPAR by fatty acids is limited. The objective of this study was to expand our understanding on modulation of bovine PPAR by FA, both when used individually and in combination, in an immortalized cell culture model of bovine liver. Of the 10 FA included in the study, the greatest activation of the PPAR reporter was detected with saturated FA C12:0, C16:0, and C18:0, as well as phytanic acid, and the unsaturated FA C16:1 and C18:1. When supplemented in mixtures of 2 FA, the most effective combination was C12:0 + C16:0, while in mixtures of 3 FA, the greatest activation was caused by combinations of C12:0 with C16:0 and either C18:0, C16:1, or C18:1. Some mixtures display a synergistic effect that leads to PPAR activation greater than the sum of their parts, which may be explained by structural dynamics within the PPAR ligand-binding pocket. Our results provide fundamental information for the development of tailored dietary plans that focus on the use of FA mixtures for nutrigenomic purposes.

## 1. Introduction

Over the last century, lipids have gained significant traction as an energy-dense dietary supplement in dairy nutrition. Sources of animal (such as tallow) or vegetable fat (soybean, palm oil, linseed, and the like) have been included in dairy cattle rations, with generally favorable results: Fats are viable sources of energy and can be used, in moderate quantities, to increase caloric availability for the animal, while generally preserving fat digestibility, and without considerable adverse effects on rumen fermentation and health, especially if saturated [1,2,3,4,5].

As outlined above, the role of dietary fat as a source of energy for dairy cows is well understood; however, significantly less emphasis has been placed on the impact of dietary fats on the transcriptome. In a recently published review written by the authors of this manuscript [6], we outlined the many fates of dietary fatty acids (FA) and their catabolism, as well as their ability to directly impact the gene expression landscape, as a function of their nutrigenomic properties. In this context, the role of peroxisome proliferator-activated receptors (PPAR) is fundamental.

PPAR isotypes are transcription factors that belongs to the nuclear receptor superfamily. The three currently known and characterized PPAR isotypes have variable expression patterns in ruminants, with PPARγ displaying high abundance in the adipose tissue, PPARα being predominant in the liver, kidney, and jejunum, and PPARδ following a more widespread expression pattern [7]. While functionally similar, all three PPAR isotypes share a common modus operandi: Upon locating a suitable intracellular ligand, the ligand-binding domain interacts with it and causes a conformational change in the receptor. PPAR then forms a heterodimer with the retinoid X receptor (RXR), another transcription factor of the same family, and, as the PPAR-RXR complex, locates a 13-nucleotide response element known as the PPAR response element (PPRE) in the promoter region of target genes, recruits the transcriptional machinery, and induces transcription or repression of said target genes [8]. Several productive and metabolic processes in ruminants are driven by genes regulated by PPAR, including the regulation of hepatic fatty acid import and catabolism [7] and insulin sensitivity in the adipose tissue [9], the synthesis of milk fat [10], and the modulation of inflammation and the immune response [7]. As such, PPAR are very relevant in developing nutrigenomic applications for bovines.

A vast amount of studies have identified FA and FA metabolites as natural ligands of PPAR in monogastrics [11]. While this should apply to ruminants as well, at least on a general level, conformational qualities of each PPAR isotype’s ligand-binding pocket are such that the degree of potency and specificity must be determined experimentally per each species. To date, several studies have identified a number of saturated and unsaturated long chain FA (LCFA) as PPAR agonists, using in vitro models of bovine mammary [12] and kidney [13] cells. However, said studies rely primarily on quantifying expression of genes known to be PPAR targets in ruminants. While this strategy is sound on a general level, functional differences between ruminant and monogastric PPAR isotypes do not guarantee that findings obtained in the latter will apply to the former [7]. As such, the use of a gene reporter encoding a measurable marker (such as a fluorescent or bioluminescent protein) driven by a PPRE is a much more reliable method, as it quantifies binding of PPAR to the reporter, and thus provides a directly proportional measurement of its activity. Using said approach, we recently demonstrated a direct, dose–response relationship between serum non-esterified fatty acids (NEFA) in transition dairy cows and PPAR activation [14]. In that study, we also observed activation of PPAR in response to palmitate, a saturated fatty acid, using immortalized bovine mammary epithelial cells [14].

Physiologically, cells are exposed to a varied mixture of FA: At any given time, more than 30 FA are circulating in bovine plasma [15]. Interestingly, ruminant PPAR appears to be activated primarily in response to long chain saturated FA [13,16], rather than unsaturated FA. Further, it has been demonstrated in monogastrics that PPARγ can bind simultaneously two fatty acids [17]. It follows that the study of the effect of FA on the transcriptome and, particularly to PPAR, should consider the presence of multiple fatty acids.

Nutrigenomics-based approaches to dairy nutrition require large amounts of accurate and focused information on the potency and specificity of each putative ingredient. To date, and to our knowledge, no such studies exist regarding the role of FA on PPAR activation in dairy cows. Thus, the objective of this study was to test a wide range of saturated and unsaturated FA on bovine cells, both individually and in combination, and to determine their ability to activate PPAR. We hypothesized that the greatest degree of PPAR activation will be measured in response to saturated FA, both when used individually and in combination.

## 2. Materials and Methods

### 2.1. Fatty Acids

Octanoic acid (C8:0; cat# A11149), decanoic acid (C10:0; cat# A14788), dodecanoic acid (C12:0; cat# 42038), and stearic acid (C18:0; cat# A12244) were purchased from Alpha Aesar (Haverhill, MA, USA). Oleic acid (C18:1; cat# BML-FA045-0010) and linoleic acid (C18:2; cat# BML-FA014-1000) were obtained from Enzo Biochem (Farmingdale, NY, USA). Myristic acid (C14:0; cat# 156961000) and palmitoleic acid (C16:1; cat# AC376912500) were purchased from Acros Organics (Thermo Fisher Scientific, Waltham, MA, USA). Palmitic acid (C16:0; cat# 100905) was from MP Biomedicals (Santa Ana, CA, USA) and phytanic acid (C20 BCFA; cat# 90360) was from Cayman Chemical Company (Ann Arbor, MI, USA). All FA except oleic acid and phytanic acid (which were in liquid form) were dissolved in DMSO (Cat# D2650-5X10ML, Sigma–Aldrich, St. Louis, MO, USA) at the maximum reported solubility for each. All resulting stock solutions were maintained at −20 °C until needed.

### 2.2. Cell Culture and Treatments

Immortalized bovine fetal hepatocytes (BFH-12, Cellosaurus accession CVCL_JQ51) were purchased from Dr. Herber Fuhrmann (University of Leipzig, Leipzig, Germany) and were grown in vented T75 flasks (cat# 25-209, Olympus Plastics, Genesee Scientific, San Diego, CA, USA) in Williams’ E Medium (cat# A1217601, Gibco, Thermo Fisher Scientific, Waltham, MA, USA) containing 5% FBS, 1% penicillin/streptomycin, 1% GlutaMAX supplement (cat# 35050061, Gibco), 100 nM of dexamethasone (cat# AAA17590, Alfa Aesar, Haverhill, MA, USA), and 0.2 U/mL of insulin (cat# I6634, Sigma–Aldrich), at 37 °C, with 5% CO_2_. Culture media choices and environmental conditions reflect the protocols developed in the original publication [18]. Cells were cultured for at least 3 passages before the beginning of each experiment.

### 2.3. Plasmid, Transfection and Luciferase Assay

PPAR activity was assayed as previously described [14]. The reporter PPRE X3-TK-luc (Addgene plasmid # 1015) was used as a measure of PPAR activation, while pRL-TK (cat# E2231, Promega, Madison, WI, USA), encoding *Renilla*
*reniformis* guided by a mini-TK promoter, was used as to account for experimental variation. Cells were transferred to white 384-well plates (cat# 781098, Greiner Bio-One North America Inc., Monroe, NC, USA) at a density of 4000 cells/well; 8 h post-passage, the plasmid mixture (40:1 ratio luciferase: *Renilla*) was transfected using Lipofectamine 3000 (cat# L3000015, Thermo Fisher Scientific, Haverhill, MA, USA) as previously described [19]. At 12 h post-transfection, selected fatty acids were added to each well. All fatty acids were added via a D300e Digital Dispenser (HP Inc., Palo Alto, CA, USA), using the manufacturer’s software to calculate a 10-point linear curve, with 0 as the minimum value, and 1% of the stock solution’s concentration as the maximum. All wells were normalized to the vehicle (DMSO) by the dispenser to 1% of total volume within the well. Plates were assayed at 18–20 h post-treatment using Dual-Glo^®^ Luciferase Assay System (cat# E2920, Promega, Madison, WI, USA), and read in a Synergy HTX Multi-Mode Reader (BioTek, Winooski, VT, USA). Integration time was 5 s/well, with the gain set at 200. Luminescence of the *Renilla* luciferase, guided by the mini-TK promoter (and as such, unaffected by the treatment) is commonly used as a measure of cytotoxicity [14]; as such, all PPAR activation estimates were obtained as a ratio of luciferase/*Renilla* luminescence.

### 2.4. Estimate Differential PPAR Activation between Mixtures of Fatty Acids

To evaluate the combinatorial effect of FA mixtures on PPAR, we compared the effect of mixtures of FA to that of each FA present in the mixture, when used individually as a treatment. As a result, two different hypothetical estimations of their combined potency were used:(1)A weighted average model where it is presupposed that each of the FA (at the dose that maximize PPAR activation when used alone) is in competition with the other to occupy a space in the LBD; thus, their estimated contribution to the total PPAR activation is proportional to their molarity, as greater concentration of a FA would increase the likelihood of a FA binding the LBD. Their combined activation can be estimated as:
(1)$$\mathrm{Expected}\;\mathrm{PPAR}\;\mathrm{activation}=\frac{\sum_{x=1}^{n}{\mathrm{FA}}_{x}\times µM_{x}}{\sum_{x=1}^{n}µM_{x}}$$
where FA refers to the activation of PPAR of an individual FA and µM refers to its molarity.

(2)An additive model that presupposes that the effect of combined FA is equal to the sum of the effect of each FA included in the mixture, which can be summarized as:
(2)$$\mathrm{Expected}\;\mathrm{PPAR}\;\mathrm{activation}=\overset{n}{\underset{x=1}{\sum }}{\mathrm{FA}}_{x}$$



### 2.5. Statistical Analysis

Results were analyzed using GLM of SAS (version 9.4, SAS Institute Inc., NC, USA) with the dose as main effect and replicate (n = 4) as random effect. Linear and quadratic effects were also evaluated in GLM. For the mixtures of fatty acids, difference between mixtures and individual FA contained within the mixtures were analyzed using GLM of SAS, using separate models containing each FA and all the mixtures that contain the FA in question. Significance was declared with a *p*-value < 0.05.

Comparisons between expected and actual PPAR activation was achieved by fitting a linear model such that:Actual Activation=a×Expected Activation+0
where a represents the slope of the curve, and 0 is set as the intercept.

## 3. Results

### 3.1. Dose–Response Activation of PPAR by Single Fatty Acids

We tested a large range of fatty acids in the 0–500 µM range to determine individual potency and dose at peak activation (Figure 1). Octanoic acid and C14:0 did not activate PPAR at any of the doses tested in our experimental design (*p* = 0.83 and *p* = 0.17, respectively). Two fatty acids activated PPAR significantly (C10:0 and C18:2, *p* < 0.001), though with a very moderate effect, peaking at just under 2-fold greater than vehicle control. Of the remaining fatty acids, four followed a quadratic tendency, namely C16:0, C18:0, C18:1 and C20 BCFA, with peak between 100 and 200 μM, while response to C12:0 was best described by a linear trend. Among the FA tested, C12:0 had the highest degree of PPAR activation (>4-fold at 500 μM) followed by 16:0 and C18:0 (>3-fold). As expected [14], doses of fatty acids > 100–150 μM decreased cell viability, as observed by the decrease in *Renilla* luminescence (Appendix A).

### 3.2. Activation of PPAR by Combination of Two Fatty Acids

Following the results presented in Figure 1, we generated all possible combinations of the six FA that displayed the greatest PPAR activation (namely C12:0, C16:0, C16:1, C18:0, C18:1, and C20 BCFA) and combined them in pairs at the dose that maximized PPAR activation individually (Figure 2). For example, from the results presented in Figure 1, we determined that maximum PPAR activation by C16:0 was achieved at a concentration of 158 µM; hence, mixtures of FA containing C16:0 contained 158 µM of this FA.

In most cases, the use of a combination of FA had higher activation of PPAR compared to the use of a single FA. However, combinations containing C12:0 displayed equal or lower activation of PPAR than C12:0 alone, with the exception of the C12:0 + C16:0 combination, which activated PPAR over 2-fold more than C16:0 alone, and ~1.3-fold more than C12:0 alone. Interestingly, the combination of C16:0 with two unsaturated FA, C16:1 and C18:1, almost duplicated the activation of PPAR compared to C16:0 alone. In the other FA tested, the combination of two FA had in general little effect or <2-fold further activation of PPAR compared to the use of the single FA. In summary, the strongest activation was achieved by combining C12:0 and C16:0 (~13-fold greater than vehicle control), C12:0 and C18:0 (~10-fold) and C16:0 with either C16:1 or C18:1 (~8-fold); however, only the combination of C12:0 + C16:0 had a higher activation compared to the use of C12:0 alone at 500 μM.

### 3.3. Activation of PPAR by Combination of Three Fatty Acids

We pooled the aforementioned six fatty acids in all possible combinations of three FA; however, to avoid previously-reported cytotoxic effects associated with high FA amounts in vitro [14], we reduced the concentration of each FA to one third of its molarity at peak PPAR activation (Figure 3). Different trends were observed among the various mixtures.

Among saturated FA, any mixture of FA containing C12:0 had higher activation of PPAR compared to C12:0 alone. Mixtures containing C16:0 led to greater PPAR activation only when combined with C12:0. Mixtures containing C18:0 activated PPAR to a greater degree than C18:0 alone only when C12:0 was in the mixture, except for the mixture containing C16:0 + C18:0 + C18:1 that activated PPAR >2-fold compared to C18:0 alone. Most mixtures containing C20:0 BCFA activated PPAR to an equal or lesser extent than C20:0 BCFA alone, except when mixed with C12:0 and C18:1.

Among unsaturated FA, compared to C16:1 alone, only mixtures containing C16:0 + C18:0 and C16:0 + C18:1 with C16:1 had higher PPAR activation. All mixtures of 3 FA containing C18:1 had greater PPAR activation compared to C18:1 alone, except when C20:0 BCFA was present.

Overall, the greatest effect was exerted by mixtures containing C12:0, namely C12:0 + C16:0 + C18:0 and C12:0 + C16:0 + C18:1 (both ~13-fold greater than vehicle control), as well as C12:0 + C16:1 + C18:1 (~12-fold) and C12:0 + C16:0 + C16:1 (~10-fold).

### 3.4. Estimated vs. Observed Activation of PPAR by Mixtures of Fatty Acids

The comparison of observed and estimated PPAR activation of each combination of FA according to both additive and weighted average models is shown in Figure 4. When two FA are mixed, a linear regression with a fixed intercept of 0 leads to a slope of 1.07 for the weighted average and of 0.62 for the additive model, indicating that a weighted average model is more adequate to describe the experimental results; this is confirmed by most treatment combinations achieving a near 1:1 ratio of observed:expected PPAR activation when the weighted average model was used (Figure 4A). On the other hand, for mixtures of three fatty acids, the weighted average model led to a slope of 2.10, while the additive model had a slope of 0.68, suggesting that as a trend, most mixtures had an experimental activation which was much greater than the weighted average of the individual fatty acids, and slightly lower than their sum. On the other hand, a group of six mixtures displayed combined activation greater than the sum of their parts (Figure 4B). Interestingly, all six of these combinations contained C12:0 and either C16:0, C16:1, C18:0, or C18:1 (Table 1).

## 4. Discussion

In line with our hypothesis, we found that of the panel present in the study, 5 out of 7 saturated FA significantly activated PPAR and four of them (C12:0, C16:0, C18:0, and C20:0 BCFA) were the most potent activators of PPAR among all tested FA. Of the 3 unsaturated FA that were assayed, all significantly activated PPAR but two activated PPAR to a relatively low degree (C16:1 and C18:1, <3-fold activation when used individually), while C18:2 had <2-fold activation.

### 4.1. C16:0 and C18:0 Confirmed to Be PPAR Agonists

The strong activation of bovine PPAR in vitro by C16:0 was previously reported [20,21], as well as its quadratic tendency towards PPAR activation [14]. Proof for the role of C18:0 in activating ruminant PPAR was also present in the literature [7,9], further evidenced by gene expression studies in ovine satellite cells and intramuscular preadipocytes, which detected an effect of both C16:0 and C18:0 on transcription of PPARγ and its target genes in the PPAR pathway [9,22].

### 4.2. C12:0 Is a Strong PPAR Activator in Immortalized Bovine Hepatocytes

A strong activation of the PPAR reporter by C12:0 was an unexpected finding. Conflicting reports exist on the interaction between medium-chain fatty acids (MCFA) and PPAR in monogastrics, with early research supporting only a modest effect of C12:0. Data from those studies revealed that C12:0 is about half as potent as the synthetic PPARα agonist WY-14,643 [23], and less effective in upregulating PPAR targets than C16:0 [24]. C12:0 is able to bind the human PPAR ligand-binding domain, though reportedly with less affinity than other MCFA [25]. Finally, supplementing mice with fed with a high-fat diet with coconut oil (rich in C12:0) reduces hypercholesterolemic and hypertriglyceridemic indicators, partially through activation of PPARα and PPARγ [26]. However, no reports of the effect of C12:0 on ruminant PPAR are present in the literature. Feeding calves with coconut oil increased in vitro esterification of fatty acids in liver slices compared to liver slices obtained from calves fed with beef-tallow (>45% oleic acid) and decreased FA oxidation [27]; however, FA oxidation was increased in both groups when cultivated with lauric acid. The higher esterification and lower oxidation of FA would be the opposite of what expected by an increased activation of PPARα. Contrary to this, cultivation of murine hepatic cells with lauric acid increased oxidation of fatty acids via activation of PPARα [28].

### 4.3. Activation of PPAR Is Differentially Modulated by Mixtures of FA and More Effective When Containing C12:0

Formulating fatty acid supplementation to maximize PPAR activation may rely not only on the utilization of individual fatty acids, but also on their use as combined supplements; that is the case with commercially-available fatty acid supplements designed for dairy cows, such as Megalac^®^, composed mainly of palmitic acid (~50%), stearic acid (~4%) and unsaturated fatty acids (cis-C18:1, ~35% and C18:2, ~7%), or Energy Booster 100^®^, which contains a mixture of palmitic (~28%) and stearic (~45%) acid. Thus, mixtures of FA are of greater significance for nutrigenomic applications.

The strong effect of C12:0 on PPAR was recapitulated in our results for combinations of 2 and 3 FA, where most treatments displaying the greatest activation of PPAR contained C12:0. Interestingly, C12:0 was present in the six mixtures that displayed greater activation than the sum of the individual potency of the FA that compose them, based on our hypothetical additive model. A suitable explanation for our unexpected findings may be found in the peculiar conformations adopted by the LBD upon encountering different ligands. A growing body of literature has evidenced the structural differences observed in the ligand-binding pocket of PPAR isotypes in response to different types and sizes of ligands: such is the case for differences in configuration between the synthetic agonists MEKT-21 and TIPP-703 [29], as well as pioglitazone and rosiglitazone [30]. Additionally, PPARγ can accommodate at least two [17] of the same oxidized FA, and at least three [25] of the same MCFA simultaneously. The ligand-binding pocket also exhibits unique degrees of affinity for different fatty acids, with human PPARα and PPARδ binding strongly to certain saturated (C14:0, C16:0, C18:0) and unsaturated (C16:1, C18:1, C18:2, C18:3) FA [31]. Finally, recent evidence suggests the possibility of co-binding of different ligands, with synthetic agonists displacing endogenous fatty acids towards an alternate ligand-binding site [32]. Interestingly, the same study found that a co-binding scenario, in which the LBD was presaturated with a PPAR agonist and a saturated (C9:0) or unsaturated (C18:1) FA was added, led to an increase in interaction between PPARγ and its obligatory coactivator TRAP220 in a synergistic fashion, and to a greater extent than the ligands individually. As such, our findings provide grounds for an enticing hypothesis: simultaneous binding of FA with different degrees of affinity to the PPAR ligand-binding pocket may result in peculiar LBD conformations (such as the one described above), which would translate to a degree of PPAR activation that is mathematically greater than the effect achievable with only one type of ligand. Further, our results suggest that unexpected conformations may only occur when certain combinations of fatty acids are included, as our findings evidence that in some cases the addition of a second or third FA does not increase PPAR activation.

### 4.4. Implications for Nutrigenomic Applications

The implications of our findings for nutrigenomic applications are multiple, and primary literature on the effect of FA feeding to livestock, though limited, seem to partly substantiate our discoveries. Several studies found an increase in milk production and milk fat synthesis in cows fed supplementary C16:0 [33,34]. This increase might be driven by activation of PPARγ by C16:0, since strong evidence exists of a major role of PPARγ in milk fat synthesis [35,36]. Likewise, genomic association studies have found an association between overrepresentation of the PPAR pathway and milk yield [37], as well as an increase in PPAR signaling at peak lactation [38,39]. It is therefore a possibility that the increase in milk yield and milk fat percentage in cows supplemented with C16:0 is due to increased activation of PPAR. However, our prior results suggest that C16:0 activates PPARα and PPARδ, but not PPARγ [14]. There is no evidence that PPARα and PPARδ affect milk fat synthesis in bovine. In sows, use of clofibrate (a PPARα agonist) did not increase milk fat or gains of the litters [40]. Whether C12:0 is a PPARγ agonist remains to be determined; however, assuming that C12:0 could activate PPARγ in vivo, this would imply that supplementing cows with C12:0 would produce an increase in milk fat comparable to that exerted by C16:0, if not superior. That seems to indeed be the case in prior studies, where feeding cows with coconut oil (about 50% C12:0) provides the same milk and milk fat yield as Megalac^®^, which is composed mostly of palmitic acid (~50%) [41]; supplementation of C12:0 alone, or coconut oil, are responsible for the same milk and milk fat yield as supplementation of C18:0, fed at equal amounts [42]. Finally, though one study found that C12:0 alone causes a decrease in both milk yield and milk fat percentage, when compared to C18:0 or C14:0, this reduction was primarily due to a decrease in dry matter intake (DMI), as DMI-adjusted yields were not significantly different [43].

All of the reference studies explore the effect of a single FA in the diet; however, as our results have shown, mixtures of fatty acids appear to be most effective in activating PPAR in the bovine liver, at least in our in vitro model. Expanding on the role of FA mixtures on PPAR modulation through dietary means would require systematic determination of optimal doses and mixtures in vitro (as is presented in this manuscript), followed by in vivo supplementation of the mixtures at equimolar concentrations, and compared to a non-supplemented control group.

## 5. Conclusions

Our results demonstrate activation of PPAR by various FA in immortalized cell culture model of bovine liver, both when supplied individually and in combination. The surprising synergy observed when combining two or three fatty acids suggests a unique mechanism of PPAR activation, possibly relying on peculiar implications of individual affinities and structural changes in the PPAR ligand-binding pocket. Data-driven FA supplementation in livestock requires systematic determination of individual and combinatorial effects in vitro before it can be tested in live animals; as such, the findings reported in this manuscript provide invaluable information to the development of nutrigenomic-based fatty acid feeding programs.

## 6. Limitations of the Study

(1)As mentioned in the discussion, relative affinity of each of the FA in our study could determine a prevalence in its binding to the LBD. Availability of such information would allow to truly test our hypothetical models of weighted average and additive effect; however, no studies on the affinity of FA to the bovine PPAR LBD are available in the literature;(2)Both the activity of the PPAR reporter and the activation of PPAR itself are variable on a time scale ([44] is an example of this). Consequently, the fact that the luciferase protocol requires lysis of the cells to obtain a reading does not allow repeated measures over time, and limits the scope of our results to the 18–24 h incubation period. This may explain discrepancies between the results presented in Figure 2 and Figure 3, where rates or PPAR activation in terms of fold change vs. vehicle control are greater than what is presented in Figure 1, as the plate which corresponds to Figure 2 and Figure 3 was incubated for ~2 h longer than that of Figure 1. However, the fact that results from the mixtures all come from the same cell culture plate, combined with the observation that relative differences between fatty acids within the same plates are maintained, suggests that the results from the mixtures can be considered reliable, and biologically relevant within the scope of the comparisons;(3)The FA supplemented in this study were diluted in culture media, and are thus more suited to mimic a local, concentrated release rather than physiological conditions. Utilizing albumin-bound FA would be more representative of a physiological state; however, it is unclear whether physiological mechanisms for the uptake of albumin-bound FA (discussed in [6]) would be maintained in a cell culture model. Ultimately, our approach provides the greatest degree of response, and as such allows to determine individual differences with greater precision;(4)BFH-12 are hepatocytes isolated and established from bovine fetuses [18] and are likely not fully representative of adult bovine hepatocytes, since the liver of calves is still maturing after few weeks post-birth [45]. However, the model should be reliable in determining the relative difference in activation and the additive effects of various FA on PPAR activation. While our model is far from perfect, its flexibility allowed us to try a vast number of treatments and combination, which is most suited to the broad scope of our study;(5)Our interpretation of the results is based on the assumption that activation of PPAR will lead to upregulation of canonical target genes which, as detailed in the introduction, are mostly involved in lipid metabolism; however, we did not measure expression of PPAR targets within the cells, which limits the ability to conclude about biological effects of the observed activation. As previously argued [9], one major limitation on using target genes to assess PPAR activation is the lack of specific bovine PPAR targets. Future endeavors must utilize the results highlighted within this manuscript as a starting point to determine response of PPAR target in models of bovine liver, and other tissues.

## Figures and Tables

**Figure 1 genes-12-01283-f001:**
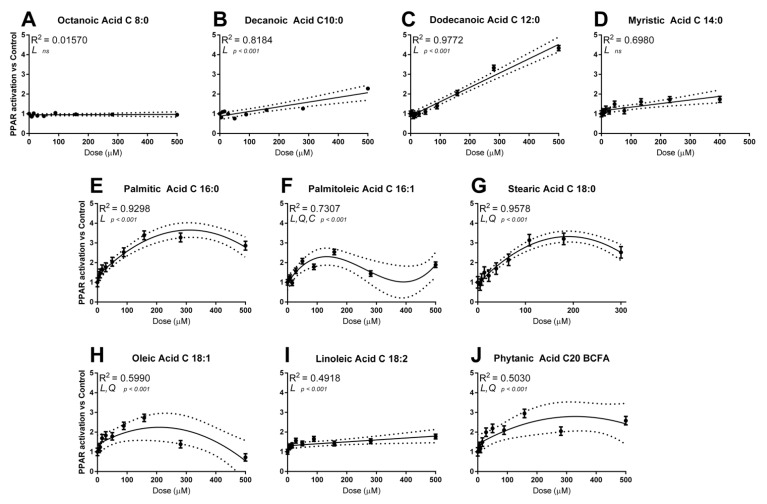
Activation of the PPAR reporter at specified doses of fatty acids octanoic acid (**A**), decanoic acid (**B**), dodecanoic acid (**C**), myristic acid (**D**), palmitic acid (**E**), palmitoleic acid (**F**), stearic acid (**G**), oleic acid (**H**), linoleic acid (**I**), and phytanic acid (**J**). Shown in each graph: significance level of the model (L = linear, Q = quadratic, C = cubic) and adjusted R^2^. Interpolations are presented as trend lines surrounded by a 95% confidence interval (dotted lines).

**Figure 2 genes-12-01283-f002:**
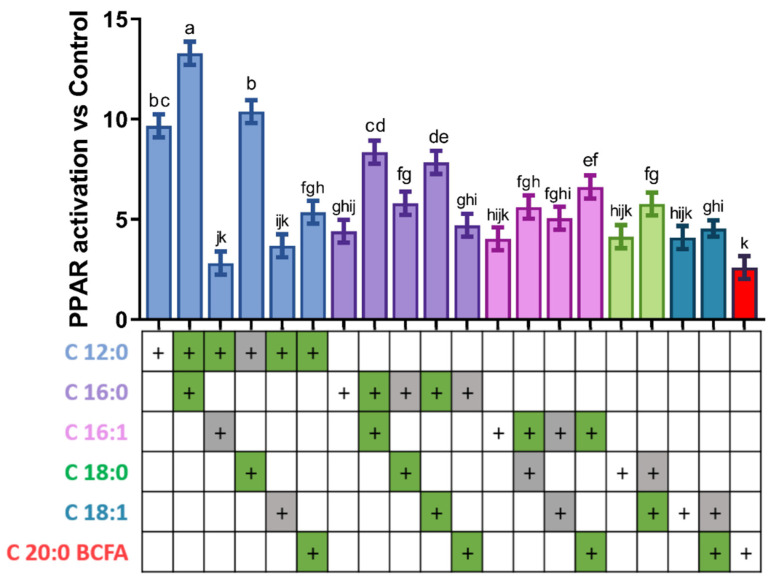
PPAR activation by combinations of two fatty acids. Compound doses are: C12:0, 500; C16:0, 158; C16:1, 158; C18:0, 190; C18:1, 158; C20:0 BCFA, 158 µM. Dissimilar letters above bars refer to statistically significant (*p* < 0.05) difference between each treatment when all pairwise comparisons are considered; green cells below the graph indicate significant differences (*p* < 0.05) in PPAR activation between the mixture and the individual fatty acid within the mixture reported on the left, when considering pairwise comparisons between the various mixture containing the fatty acid and the individual fatty acid, while gray cells indicate no significant differences (*p* > 0.05).

**Figure 3 genes-12-01283-f003:**
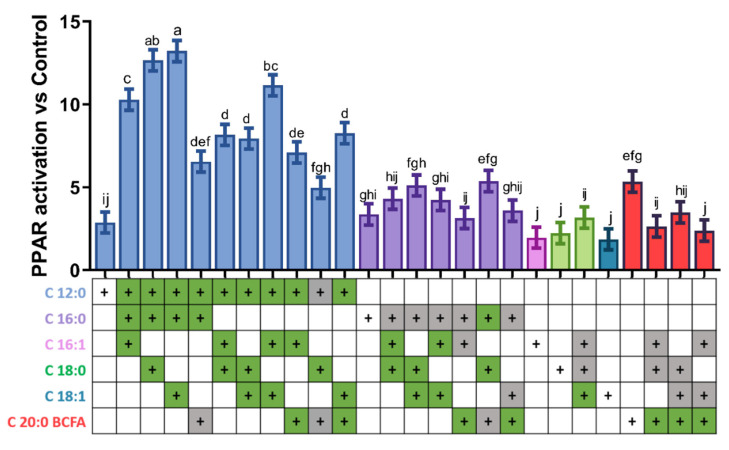
PPAR activation by combinations of three fatty acids. Compound doses are: C12:0, 167; C16:0, 53; C16:1, 53; C18:0, 63; C18:1, 53; C20:0 BCFA, 53 µM. Dissimilar letters above bars refer to statistically significant (*p* < 0.05) difference between each treatment when all pairwise comparisons are considered; green cells below the graph indicate significant differences (*p* < 0.05) in PPAR activation between the mixture and the individual fatty acid within the mixture reported on the left, when considering pairwise comparisons between the various mixture containing the fatty acid and the individual fatty acid, while gray cells indicate no significant differences (*p* > 0.05).

**Figure 4 genes-12-01283-f004:**
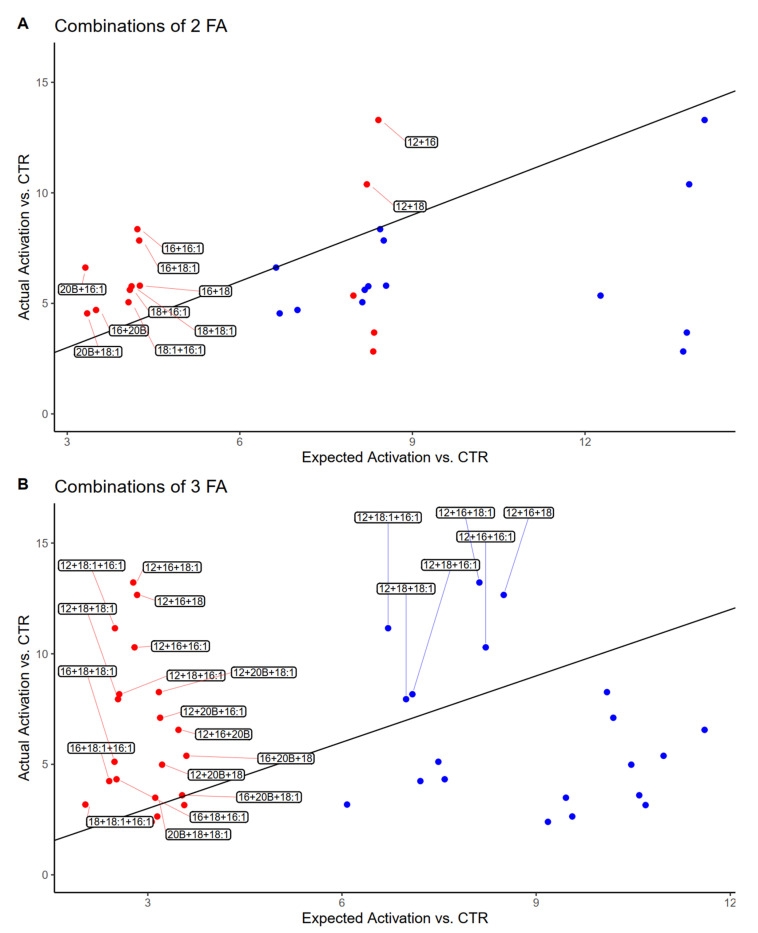
Dot plot comparing mean of experimentally measured PPAR activation by combinations of two (**A**) or three (**B**) fatty acids, when compared to the hypothetical weighted average model (red dots) and additive model (blue dots). Diagonal line represents threshold where ratio of actual/expected activation is equal to 1. Labeled in the figure: mixtures with an actual/expected activation ratio greater than one. (12 = dodecanoic acid; 16 = palmitic acid; 16:1 = palmitoleic acid; 18 = stearic acid; 18:1 = oleic acid; 20B = phytanic acid).

**Table 1 genes-12-01283-t001:** Measured modulation of the PPAR reporter as fold-change vs. control—i.e., not FA treatment—in response to mixtures of three FA containing C12:0, compared to expected activation based on the weighted average and additive model. %Δ: percentage increase in measured PPAR modulation over expected value.

Combination	Measured	Expected			
		Weighted Aver.	%Δ	Additive	%Δ
12:0 + 16:0 + 16:1	10.3	2.8	268	8.2	25
12:0 + 16:0 + 18:1	13.2	2.8	376	8.1	63
12:0 + 16:0 + 18:0	12.7	2.8	346	8.5	49
12:0 + 16:1 + 18:0	8.2	2.6	219	7.1	15
12:0 + 16:1 + 18:1	11.2	2.5	348	6.7	66
12:0 + 18:0 + 18:1	7.9	2.5	213	7.0	14

## Data Availability

The raw data supporting the conclusions of this article will be made available by the authors upon request, without undue reservation.

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
