# Peer review of "When Two plus Two Is More than Four: Evidence for a Synergistic Effect of Fatty Acids on Peroxisome Proliferator—Activated Receptor Activity in a Bovine Hepatic Model"

_genes, 2021, doi:10.3390/genes12081283_

Round 1
Reviewer 1 Report
The study was done to evaluate the supplementation of different fatty acids in different combinations on PPAR activity in bovine hepatic cell model. As the authors point out, most of the earlier studies on the role of PPARs had assayed the expression of genes known to be regulated by PPARs. The authors evaluate the PPAR activity in this study. The results suggest an important role for C12:0 fatty acid in addition to C16:0, among others in activating PPARs in bovine cell model.
Additional information on the following is necessary for better clarity to the manuscript.
- What was the status of the cells after the Fatty acid treatment? A higher dose of fatty acids tends to have toxic effects on the cells. It’s possible the cell number would be different after the treatment. How did you normalize the cell number after the treatment??
- When the combinations of fatty acids are tested, how can you be sure that the fatty acids bind to the receptor in the same proportion as the one added. For eg., If you added FA1 and FA2 in 50% each, how can you be sure that they are binding equally to elicit the response. Would it be possible that there could be competition among the FA to bind to the receptor depending on the type of FAs or is it a random occurrence?
- Line 183. It is unclear how the doses for the combinations were decided. Please elaborate with one example for better comprehension for the readers. Would the responses be different if the combinations of the same fatty acids are in different ratio. Why?
- Did you keep the dose of total Fatty acid the same irrespective of the concentrations of different fatty acids. Would the responses be different if the total amount of fatty acid supplementation differs when different combinations of fatty acids are used?
- Did you measure the expression of genes that are downstream to PPARs. That information would have supported the responses to the PPAR activation by some of the fatty acids. The responses of the fatty acids in different tissues tend to be different eg., liver vs Mammary. If the responses are different, the extrapolation of the results from this to increased milk fat in dairy cows as mentioned in the discussion may have to be done with caution.
Author Response
- What was the status of the cells after the Fatty acid treatment? A higher dose of fatty acids tends to have toxic effects on the cells. It’s possible the cell number would be different after the treatment. How did you normalize the cell number after the treatment??
AU: Of the two plasmids that were cotransfected, 3xPPRE-TK-luc is a luciferase driven by PPAR activation, while pRL-TK is a renilla luciferase driven by a mini-TK promoter, and is a direct measurement of both cellular viability, cells number, and transfection efficiency. The results are obtained by first generating a ratio of luciferase/renilla, and then scaling the resulting number to the untreated control to calculate fold induction. We added a section in the methods to clarify (Lines 132-136). Further, we added a supplementary figure (Supplementary Figure 1) showing renilla luminescence in response to the treatment, as it has been used successfully in the past to estimate cell viability as indicated in the manuscript. Despite a decrease in cell viability with dose >100-150 μM, the normalization using Renilla would account for decrease in cell number/activity; thus, the relative activation of PPAR is correct.
- When the combinations of fatty acids are tested, how can you be sure that the fatty acids bind to the receptor in the same proportion as the one added. For eg., If you added FA1 and FA2 in 50% each, how can you be sure that they are binding equally to elicit the response. Would it be possible that there could be competition among the FA to bind to the receptor depending on the type of FAs or is it a random occurrence?
AU: That is a good question, one that we unfortunately cannot answer. As described in the discussion and limitation sections of the manuscript, a plausible scenario indicating competition of the FA within the mixture would imply that the overall binding complexes would depend on the individual binding affinity of each FA to the ligand-binding pocket. As noted in (1) and discussed in lines 325-339, the ligand-binding domain of human PPAR has greater affinity to certain FA than for others; for example, if a mixture of C16 (high affinity) and C8 (rel. low affinity) are supplied at equal molarity, it is likely that more PPAR molecules will be bound to C16 than C8. However, this presents several issues: 1) human and bovine PPAR ligand-binding pockets are considerably different, which implies that whatever (limited) information is present on binding affinities of individual FA for humans will not apply for bovines; 2) obtaining such data for bovines is well outside our capabilities, and 3) it still would not explain some of our results, as synergistic effects would suggest that all the supplied FA in the mixture are binding the receptor.
- Line 183. It is unclear how the doses for the combinations were decided. Please elaborate with one example for better comprehension for the readers. Would the responses be different if the combinations of the same fatty acids are in different ratio. Why?
AU: Thank you for the observation. The doses for each FA were chosen based on the results in Figure 1, namely each FA was supplied at the concentration at which it was observed to activate PPAR to the greatest extent, for example 158 uM for C16:0, 500 uM for C12:0, etc. An additional sentence was added to specify it (Lines 188-191). Further, the concentrations are listed on the label of each figure containing mixtures of FA (Fig. 2 and 3). As for the second part of the question, we expect to obtain different responses if the ratios are altered; however, selecting the dose that maximized PPAR activation when used individually FA allowed each FA to express its peak effect within the mixture.
- Did you keep the dose of total Fatty acid the same irrespective of the concentrations of different fatty acids. Would the responses be different if the total amount of fatty acid supplementation differs when different combinations of fatty acids are used?
AU: Due to the nature of how our mixtures were calculated (each FA at its peak activation), it was not possible to supply all mixtures at the same dose; however, the discussion of our results only takes into account comparisons between the mixture and each fatty acid within the mixture, and limited comparisons are presented between different mixtures. As an example: if mixture M1 is made of fatty acids F1 and F2, we would compare activation of PPAR by M1 with activation by F1 and F2 individually, as well as the sum and weighted average of F1 and F2; comparisons between M1 and other mixtures were relatively ignored due to this.
- Did you measure the expression of genes that are downstream to PPARs. That information would have supported the responses to the PPAR activation by some of the fatty acids. The responses of the fatty acids in different tissues tend to be different eg., liver vs Mammary. If the responses are different, the extrapolation of the results from this to increased milk fat in dairy cows as mentioned in the discussion may have to be done with caution.
AU: We did not, as our primary focus was to assay direct activation of PPAR; further, it is not entirely clear what “canonical” downstream PPAR targets in monogastrics apply to bovines as well. The reviewer does bring up a good point; as such, we have added a paragraph on the limitations section regarding not having measured PPAR targets (paragraph 5, lines 416-422)
Reviewer 2 Report
Isotypes of PPAR regulate important metabolic processes in both monogastric and ruminant animals, including the metabolism of fatty acids (FA), the production of milk fat, and the immune response; however, information on the modulation of bovine PPAR by fatty acids is limited. The objective of this study was to expand the understanding on modulation of bovine PPAR by FA, both when used individually and in combination, in an immortalized cell culture model of bovine liver. The findings reported in this manuscript provide the valuable information to the development of nutrigenomic-based fatty acid feeding programs.
Concerns
1 When multiple fatty acids are used in combination, why the amount of C12:0 is different from that of other fatty acids?
2 The results revealed that the combination of two fatty acids, when they combined with C12:0 that performed the strongest activation, but the combined use of C16:1 and C12:0 is not as effective as C16:1 alone. At the same time, no matter what kind of fatty acid, when used in combination with C16:0, there are no significant impairs. Thus, the combination of C12:0 and C16:0 enhanced 2-fold PPAR activation compared to C16:0 alone, this might be attributed to C16:0 but not C12:0.
Author Response
- When multiple fatty acids are used in combination, why the amount of C12:0 is different from that of other fatty acids?
AU: The concentrations of the mixtures were determined based on the maximal activation when FA were used individually. For example: the maximum activation of PPAR by C16:0 was at 158 uM, by C18:0 was 190 uM, and by C12:0 was 500 uM (the activation of PPAR increased linearly). So, a combination of these 3 would have 158 uM C16:0 + 190 uM C18:0 + 500 uM C12:0. See Figure 1 for doses at maximum activation. Additionally, we have added a section explaining in further detail how the doses were obtained (lines 188-191)
- The results revealed that the combination of two fatty acids, when they combined with C12:0 that performed the strongest activation, but the combined use of C16:1 and C12:0 is not as effective as C16:1 alone. At the same time, no matter what kind of fatty acid, when used in combination with C16:0, there are no significant impairs. Thus, the combination of C12:0 and C16:0 enhanced 2-fold PPAR activation compared to C16:0 alone, this might be attributed to C16:0 but not C12:0.
AU: The combination C12:0 + C16:1 activates PPAR to the same extent as C16:1 (same means group, no significant differences between the two treatments). In all of the analyzed mixtures, PPAR was activated at least as strongly as the FA (within that mixture) with the lowest effect; that is to say: if FA1 activates 4 fold, and FA2 activates 2 fold, the combination is never lower than 2 fold. The same concept applies to the mixtures of C12:0+C16:0, the sum of which corresponds almost exactly to the effect of C12:0+the effect of C16:0.